# Air Pollution Exposure Monitoring among Pregnant Women with and without Asthma

**DOI:** 10.3390/ijerph17134888

**Published:** 2020-07-07

**Authors:** Sandie Ha, Carrie Nobles, Jenna Kanner, Seth Sherman, Seung-Hyun Cho, Neil Perkins, Andrew Williams, William Grobman, Joseph Biggio, Akila Subramaniam, Marion Ouidir, Zhen Chen, Pauline Mendola

**Affiliations:** 1Department of Public Health, Health Sciences Research Institute, College of Social Sciences, Humanities, and Arts, University of California, Merced, CA 95343, USA; 2Division of Intramural Population Health Research, Eunice Kennedy Shriver National Institute of Child Health and Human Development, Bethesda, MD 20892, USA; cnobles@umass.edu (C.N.); jenna.kanner@gmail.com (J.K.); perkinsn@mail.nih.gov (N.P.); marion.ouidir@nih.gov (M.O.); chenzhe@mail.nih.gov (Z.C.); pauline.mendola@nih.gov (P.M.); 3The Emmes Company, Rockville, MD 20850, USA; ssherman@emmes.com; 4RTI International, Research Triangle Park, NC 27709, USA; scho@rti.org; 5School of Medicine & Health Sciences, University of North Dakota, Grand Forks, ND 58202, USA; andrew.d.williams@und.edu; 6Feinberg School of Medicine, Northwestern University, Chicago, IL 60611, USA; w-grobman@northwestern.edu; 7Ochsner Health System, New Orleans, LA 70115, USA; joseph.biggio@ochsner.org; 8Department of Obstetrics & Gynecology, University of Alabama at Birmingham, Birmingham, AL 35294, USA; asubramaniam@uabmc.edu

**Keywords:** air pollution, pregnancy, asthma, personal air monitoring, exposure assessment, monitoring compliance

## Abstract

*Background*: We monitored exposure to fine particulates (PM_2.5_), ozone, nitrogen dioxide (NO_2_), and ambient temperature for pregnant women with and without asthma. *Methods*: Women (*n* = 40) from the Breathe—Well-Being, Environment, Lifestyle, and Lung Function Study (2015–2018) were enrolled during pregnancy and monitored for 2–4 days. Daily pollutants were measured using personal air monitors, indoor air monitors, and nearest Environmental Protection Agency’s stationary monitors based on GPS tracking and home address. *Results*: Personal-monitor measurements of PM_2.5_, ozone, and NO_2_ did not vary by asthma status but exposure profiles significantly differed by assessment methods. EPA stationary monitor-based methods appeared to underestimate PM_2.5_ and temperature exposure and overestimate ozone and NO_2_ exposure. Higher indoor-monitored PM_2.5_ exposures were associated with smoking and the use of gas appliances. The proportion of waking-time during which personal monitors were worn was ~56%. Lower compliance was associated with exercise, smoking, being around a smoker, and the use of a prescription drug. *Conclusions*: Exposure did not vary by asthma status but was influenced by daily activities and assessment methods. Personal monitors may better capture exposures but non-compliance merits attention. Meanwhile, larger monitoring studies are warranted to further understand exposure profiles and the health effects of air pollution during pregnancy.

## 1. Introduction

Asthma is characterized by inflammation of the airway, resulting in swelling that can adversely affect breathing. It is a common chronic disease affecting approximately 8% of pregnant women and is associated with pregnancy complications [1]. The course of asthma changes throughout pregnancy, with approximately one-third of the affected women having improved symptoms, one-third having worsened symptoms, and one-third having no change in symptoms, although women with more severe asthma at baseline have been shown to have a greater chance of exacerbation during pregnancy [2,3]. The causes of these changes are unclear but they may be influenced by infection, gastroesophageal reflux, changes in medication regimen, and smoking [2].

Ubiquitous environmental exposures, such as air pollution, increase the risk of asthma exacerbation in the general population through oxidative stress and inflammatory responses [4,5]. However, the effects of common air pollutants on asthma among pregnant women have not been well investigated. Asthma has been associated with higher risk for preeclampsia [6] and preterm birth [7] as well as neonatal respiratory complications associated with air pollutants but not all adverse outcomes of pregnancy vary by asthma status [8,9]. Regardless of asthma status, air pollution exposure appears to increase the risk for adverse pregnancy outcomes including pregnancy loss, stillbirth, preterm delivery and infant growth restriction [10,11,12,13,14]. Due to feasibility constraints, existing studies on the health effects of air pollution often rely on indirect estimates of air pollution exposure. For example, many earlier studies rely on measurements at the closest stationary air monitor to a location of interest such as a person’s residential address [15]. This method is simple and easy to implement but is highly subject to exposure misclassification as it cannot adequately capture small spatial and temporal variability. In other words, it assumes that a person is exposed to the same levels of air pollution recorded at the closest stationary monitor to their home, which could be tens of miles away.

Complex mathematical models have also been developed to allow flexibility in incorporating environmental parameters (e.g., weather) that influence the spatiotemporal distribution of air pollution [10,15]. While more sophisticated models can provide more spatiotemporally accurate estimates, they are still indirect estimates, and are likely influenced by unverifiable model assumptions. Additionally, these models still treat exposure as a point source and cannot account for daily activity patterns. Global positioning system (GPS)-based technologies have also been used to improve estimation by capturing local mobility throughout the day [16].

Since people are mobile throughout the day, the best method for assessment of air pollution exposure is personal air monitoring, which requires participants to wear air monitors throughout the day. This method captures what a person breathes more accurately and can improve the ability to determine the true effects of air pollution on health. However, because personal monitoring is expensive and burdensome to the participants, it is difficult to implement. Since most of the population spend the majority of their time indoors, studies have also used indoor monitors to estimate personal exposure [17]. This method is thought to be more accurate than the indirect methods and can alleviate some of the daily burden of personal monitoring for study participants, thus improving compliance. Few studies have simultaneously implemented and compared these exposure assessment methods.

Pregnancy is a relatively short window of susceptibility, which presents a unique opportunity to assess air pollution exposures using personal monitoring together with other assessment methods. Whereas wearable monitors may improve the quantification of personal exposure, very few studies have simultaneously compared their feasibility with other common and less expensive assessment methods for pregnancy studies. The purpose of this study was to characterize air pollution exposures among pregnant women with and without asthma using four methods based on (1) the closest air monitor to residence, (2) the closest air monitor to GPS-monitored locations throughout the day, (3) indoor air monitors, and (4) personal portable air monitors. Furthermore, we also explored daily activities that can affect air pollution exposures and investigated factors that may influence the proportion of time participants actively wear the personal monitor during the monitoring period (i.e., personal monitoring compliance).

## 2. Methods

### 2.1. Study Sites

Our study sites include the Center for Women’s Reproductive Health at the University of Alabama, Birmingham, Alabama; and Northwestern University in Chicago, Illinois. These two sites have different environmental profiles and demographics. Chicago is a more metropolitan area and generally has higher air pollution concentrations. A comparison of the characteristics of these two areas can be found in Appendix A.

### 2.2. Data and Participants

The Breathe—Well-Being, Environment, Lifestyle, and Lung Function Study (B-WELL-MOM) is a multi-center prospective cohort study (2015–2018) that aims to investigate factors associated with poor asthma control during pregnancy, as well as the basic immunology of asthma in pregnancy. Pregnant women were eligible for enrollment if they were ≥18 years old; <15 weeks of gestation with a single gestation; an English or Spanish speaker; without a diagnosis of morbidities associated with immunologic alterations (e.g., HIV, multiple sclerosis, rheumatoid arthritis or mixed connective tissue disease); not expecting to terminate pregnancy; and planning to deliver at one of the study hospitals. To be classified in the active asthma study group, women with a history of asthma must have had symptoms or used prescription medication for asthma in the year prior to pregnancy. To be classified in the no asthma study group, women must have no history of an asthma diagnosis and no current asthma. The original cohort enrolled a total of 418 women including 147 with well-controlled asthma, 164 with poorly controlled asthma, and 107 with no asthma, defined by the American College of Obstetrics and Gynecology’s classification of asthma severity during pregnancy based on medical records and self-reports [18]. Pregnant women completed a clinical interview with centrally trained interviewers, performed a 24 hr dietary recall, and provided biological specimen during four study visits at gestational weeks <15, 20–22, 30–32, and at four months post-partum. At delivery, medical chart abstraction was used to obtain detailed clinical information. In addition, infant anthropometric measures were obtained at birth and at the post-partum visit. Women also completed daily diary and in-home assessments of respiratory functions (peak flow measurements and exhaled nitrous oxide) during pregnancy (Figure 1). The present analysis includes a subset of 40 pregnant women enrolled in the B-WELL-MOM study who agreed to participate in the air-monitoring sub-study.

### 2.3. Air Pollution Assessment

During study visits 2 or 3, pregnant women were asked to participate in the 4 day air-monitoring sub-study and were given wearable real-time sensors and passive badges to collect personal exposures and indoor air quality measurements. RTI MicroPEM^TM^ wearable real-time sensors were used to collect PM_2.5_ data and filter samples; Cairpol CairClip (Altech) real-time sensors for NO_2_; and two Ogawa passive badges for O_3_ (Appendix A) [19]. These sensors have light weights (230, 55, and 18 g, respectively) and compact sizes to be used as low-burden personal monitors for pregnant women. MicroPEM, CairClip and Ogawa badges were the best available technologies for wearable sensors when those real-time sensors were chosen in 2015 [20,21,22]. Prior to each MicroPEM deployment, each individual MicroPEM’s flow rate and baseline response were calibrated [23]. The real-time PM_2.5_ concentration data were post-corrected using the corresponding gravimetrically determined concentration measured by the filter samples (gold-standard measurement). Real-time sensor-operating parameters such as inlet pressure drop, orifice pressure drop, and flow rate were examined during the validity check. The MicroPEM units were set to run for at least 5 days continuously using 30 second on/off cycling at a flow rate of 0.5 L/min. All CairClips were examined for precision through a collocation study before the air-monitoring study began. For Ogawa badges, shipping and lot blank samples were collected throughout the study to correct for potential contamination and background levels, respectively.

Participants were instructed to carry the monitors with them in a mesh pouch for 48 hours (personal assessment) and then place them in the main living quarter of their home (i.e., where people spend the majority of their time) for 48 hours (indoor assessment) (Appendix A). Real-time sensors (MicroPEM’s, CairClips) were operated continuously throughout the monitoring period, and each of two Ogawa badges was deployed to collect separate personal and indoor samples. Participants also carried a tablet device for mobility tracking, and to complete a daily diary to assess daily activities as well as self-reported symptoms. Due to privacy concerns, we restricted the resolution of the mobility measure to 100 meters (approximately the length of one football field) so that exact locations were not collected. To ensure proper sensor deployment, utility and retrieval, video instructions were made available to both study staff and participants (Appendix A). Informed consent was obtained from all participants and the study protocol was approved by Institutional Review Boards from all participating institutions.

In addition to personal and indoor monitoring, pollutants were also assessed using two common methods: stationary EPA monitor + home address, and stationary EPA monitor + GPS locations. First, participants’ home addresses were spatiotemporally linked to the nearest EPA air monitor (average distance: 11.2 km; range: 2.0–19.3 km). Daily average concentrations of PM_2.5_, NO_2_, O_3_, and temperature for each woman were estimated during the 4 days of the air-monitoring sub-study using observed measurements at the nearest EPA monitor. Second, participants’ GPS locations throughout the day (approximately every five minutes) during the monitoring period were spatiotemporally linked to the nearest EPA air monitor (average distance: 11.8 km; range: 0.4 and 47.7 km). Daily exposure to specific air pollutants during the study period was then estimated using the average location-specific concentrations observed at the nearest EPA monitors.

### 2.4. Daily Activities

During the monitoring period, pregnant women were asked to self-report daily activities through an app on their tablet, or on a written form. These activities include whether they spent time near heaters, burning trash, or smokers; used a gas range/oven/cooktop, water heater, gasoline equipment’s, or printer.

### 2.5. Monitor Compliance Assessment

The MicroPEM device assesses participant activity level using an internal 3-axis accelerometer, which was used to validate participants’ compliance of wearing monitors. Monitor compliance was operationalized as the proportion of time the monitor was worn during waking hours. This was calculated using the activity level data recorded by the accelerometer of the MicroPEM and self-reported sleeping hours, if provided. If sleeping hour information was not available, we assumed an 8 h sleep time per day.

### 2.6. Predictors of Compliance

Potential predictors of compliance were obtained using a combination of questionnaires, daily diaries, and in-home assessments. Maternal age, body mass index (BMI), and asthma status were assessed using questionnaires at baseline visit. Symptoms and daily activities were assessed using the daily diaries. Peak expiratory flow was collected in the morning and afternoon during daily in-home assessment using the Pocket Peak^®^ Mechanical Peak Flow Meter (nSpire Health, Inc., Longmont, CO, USA). Exhaled nitrous oxide was collected during in-home assessment using the NIOX Vero (Aerocrine). Women were provided with proper training on how to use these devices during their clinical visits and were provided detailed instructions. In addition, daily diaries also assessed symptoms and various daily activities during the monitoring period.

### 2.7. Statistical Analysis

Fisher’s exact test and t-test were used to compare time-invariant categorical and continuous characteristics between women with and without asthma, respectively. For time-varying factors (i.e., daily air pollution), mixed models were used to account for within-woman variations. To determine the types of activities (yes/no) associated with daily air pollution exposures, we used generalized linear mixed models. Since the distributions for pollutants concentrations were positively skewed, log transformation was used prior to modelling. The effect estimates were then converted to percent difference (and respective 95% confidence intervals) in concentrations of air pollutants associated with specific activities. To determine factors associated with waking-hour monitor-wearing time (i.e., compliance), generalized linear mixed models were used to estimate the differences in monitoring wearing time associated with factors such as baseline asthma status, age, BMI, respiratory function measures, and various daily activities and symptoms. Given that the goal of our analysis was to explore predictors of monitor-wearing time, and the low sample size, we only used univariable models for our main analyses. Models with all significant factors yielded generally consistent conclusion with univariate models, so we opted to present the results of univariable models.

## 3. Results

The analysis includes a total of 40 pregnant women who participated in the air pollution-monitoring sub-study (with 24 provided GPS data). There were 19 (47.5%) women with well-controlled asthma, 12 (30.0%) with poorly controlled asthma, and 9 (22.5%) without asthma at baseline (Table 1). Women with asthma generally enrolled in this sub-study earlier in pregnancy compared to their counterparts (gestational week 23 vs. 27, *p* = 0.06). The majority of pregnant women (72.5%) were ≥30 years old or overweight/obese (80.0%). Women actively wore the air monitors for approximately 55.7% of the monitoring time during waking hours and reported spending an average of 93 minutes outdoors per day. These characteristics did not differ by baseline asthma status (Table 1).

Average home distance from EPA air monitors ranged from 6.3 to 23.5 km depending on the types of pollutant, and was independent of baseline asthma status. Temperature monitors were typically further from participants’ home (Table 1), as temperature monitoring is not always available at all monitoring stations. GPS distance from the nearest EPA air monitor did vary by asthma status. Women with poorly controlled asthma tended to be closer to PM_2.5_ monitors (5.8 km vs. 8.1 and 6.3 km, *p* < 0.01), ozone monitors (7.2 km vs 9.3 and 8.1, *p* < 0.01) and NO_2_ monitors (4.3 km vs. 7.0 and 8.1, *p* < 0.01) compared to women with well-controlled asthma or no asthma (Table 1). This finding is consistent when excluding the one GPS participant from one of the two sites (not shown). Women with poorly controlled asthma had lower peak flow values, and higher rates of wheezing.

Overall, PM_2.5_, ozone, NO_2_, and temperature exposures among asthmatic and non-asthmatic pregnant women did not differ significantly when estimated with personal or indoor monitors (Table 2, Appendix A). However, women with asthma tend to be exposed to higher O_3_ concentrations and higher ambient temperature compared to their counterparts when estimated using EPA monitors measurements. More specifically, when exposures were estimated using GPS or home address locations + the EPA monitor method, women with poorly controlled asthma were exposed to a daily O_3_ average of 33.9 ppb compared to 28.4 ppb among well-controlled and 24.6 ppb among non-asthmatics (*p* = 0.01). Similarly, women with poorly controlled asthma were exposed to higher ambient temperature (15.9 °C vs. 12.1 and 9.8 °C, *p* = 0.01) compared to their counterparts (Table 2, Appendix A). These results are consistent even when excluding the one participant from one of the sites (Appendix A).

Pollutant estimates were significantly different across different assessment methods (Table 2, Appendix A). Compared to EPA monitor-based assessment methods, personal and indoor air monitors generally captured higher concentrations of PM_2.5_ exposures and higher ambient temperature, but lower concentrations of O_3_ and NO_2_, both of which are generally outdoors pollutants. It is also important to notice that personal and indoor monitors captured more variability, including PM_2.5_ concentrations that reached well above the EPA’s daily standard of 35 ug/m^3^ (Table 2, Appendix A) [24]. The findings are generally consistent for when stratified by study sites (Appendix A). A correlation matrix with air pollutant exposures estimated by different assessment methods is also presented in Appendix A. In general, there is strong correlation between GPS- and home-based estimation methods for all pollutants. Personally measured exposures to PM_2.5_ (r = 0.32), NO_2_ (r = 0.76), and temperature (r = 0.76) were positively associated with indoor-monitor measurements. Estimates using EPA monitors were generally not positively correlated with personal or indoor estimates, except for temperature.

Table 3 describes the percent difference in exposures associated with various daily activities. Pregnant women who used gas appliances during monitoring time (e.g., cook top or oven) had 194.6% (95% CI: 55.0–549.6%) higher indoor exposures to PM_2.5_ compared to those who did not. Those who smoked also had almost 400% (95% CI: 78.5–1293.8%) higher indoor concentrations of PM_2.5_. Generally, using a gas appliance, clothes dryer, or a printer and spending time near a heater, burning trash, water heater, or smoking were associated with higher exposures to PM_2.5_, O_3_, and NO_2_, but these associations were not statistically significant, possibly due to the low sample size.

Compliance for women in the study, defined by the percent of time air monitor was worn during waking time, was approximately 55.7%, and did not differ by asthma status (Table 4). Having a missed workday and being near a heater was associated with approximately a 20% (95% CI: 1–39%) and 42% (34–50%) increase in compliance, respectively. On the other hand, exercise (−11%, 95%CI: −19–−3%), smoking (−23%, 95% CI: −30–−16%), being around a smoker (−16%, 95% CI: −26–−6%), and using a prescription drug (−13%, 95% CI: −26–−0.4%) were associated with significantly lower waking-hour monitor-wearing time. When all significant factors were included in a single model, results were consistent, except missing a work day was no longer significant (data not shown).

## 4. Discussion

The purpose of this study was to characterize air pollution exposures among pregnant women with and without asthma using four different assessment methods, including personal and indoor sensor measurements and estimation using home address and GPS location in combination with EPA ambient monitoring station data. We also explored daily activities that may affect air pollution exposures, and factors influencing the proportion of waking time during which monitors were worn.

Our analyses suggest that when measured with personal or indoor monitors, exposures to PM_2.5_, O_3_, NO_2_, and temperature did not differ by baseline asthma status. On the contrary, when estimated by EPA monitors (both based on home address and GPS movements), women with poorly controlled asthma appeared to have significantly higher exposures to O_3_ and higher ambient temperature independent of site differences. This may suggest that women with poorly controlled asthma may live and move around in neighborhoods with higher outdoor concentrations of O_3_ and warmer climates, both of which are not captured by personal monitors given that women spent most of their time indoors. Other studies examining outdoor air pollution and neighborhood environment have also shown that people living in more polluted neighborhoods may have worse asthma morbidity [25,26,27,28]. However, these differences have not been well investigated among pregnant women and merit further attention. Studies have also shown that vulnerable subgroups of the population often live and work in areas with warmer outdoor microclimate due to socioeconomic differences including greenspace and asphalt surfaces [29,30]. It is important to note that we examined exposures based on baseline asthma status, while it is possible that women’s asthma status may change throughout pregnancy [31]. Thus, future studies should examine exposures in relation to asthma status change throughout pregnancy.

Our data also show that exposures varied significantly by assessment methods. More specifically, local outdoor monitor-based (e.g., EPA monitor) methods appeared to underestimate PM_2.5_ and ambient temperature, and overestimate O_3_ and NO_2_ exposures relative to personal and indoor monitors. The indoor–outdoor differences in PM_2.5_ and O_3_ are generally consistent with other studies in the US and around the world [32,33,34]. These differences have important implications for future studies, as they support the notion that local air monitors lack the ability to capture small spatial variations in different exposure microenvironments. Outdoor and indoor monitors also capture different sources of pollution but, since people spend most of their times indoor, indoor monitoring may have the greatest contribution to personal exposure, if compliance is reasonable. Our data also show that the GPS-based and home-based methods yielded similar estimates with high correlation, suggesting that accounting for outdoor space-time activities do not significantly influence exposure estimates [35].

Personal air monitors have the advantage of being able to capture more exposure variability, as shown in our data as well as other studies [36]. However, our data also suggest that compliance is approximately 56% of waking time, which presents an important challenge in environmental studies that involve personal monitoring. We also note that a prior study investigating personal exposure protocol compliance shows that although compliance can vary between people, it does not appear to misrepresent exposures measured longitudinally when one can reach a threshold of approximately 40% [37]. Given that women spend most of their time indoors [38], a trend that is also reflected in our data, indoor monitoring may offer a reasonable alternative to alleviate participant burden, especially considering our results which suggest that indoor estimates are very similar to measurements from personal monitors. We also observed that having a missed workday and being near a heater are associated with higher compliance, and this may suggest that women are more likely to wear the personal monitors when they are indoors.

Daily activities may influence exposures. Women who smoked and used gas appliances had significantly higher exposures to indoor fine particles, but they were also significantly less likely to wear monitors during waking hours. The differences in exposures based on these daily activities were reflected in indoor monitoring, which provides another reason indoor monitoring may be a reasonable alternative in future studies. In addition, some common daily activities, including the use of gas appliances, are associated with higher exposure to pollutants. As such, it is important to raise awareness regarding the potentially harmful effects of these pollutants and encourage strategies for minimizing exposure such as the use of vent hoods, air purifiers, and the avoidance of active/passive smoking. Meanwhile, there is a need for more comprehensive and larger studies to investigate personal exposure data together with daily diaries to identify activities that cause high exposure conditions in order to prepare effective and reasonable intervention efforts.

A study among physicians in Poland showed that only approximately one-quarter of physicians believed that their knowledge about the health effects of air pollution is sufficient, and only 3.5% answered the general knowledge questions correctly [39]. In the US, despite the well-described health effects of air pollution, the majority of physicians reported never talking to their patients about limiting exposures [40]. This suggests a clear need for tools and resources to help health care providers communicate the deleterious effects of air pollution [41], especially to vulnerable populations such as pregnant women. Meanwhile, well-designed resources for the general public are also necessary. These efforts need to ensure that the communication strategies include critical information including risk mitigation behaviors and health impacts, and that they reach vulnerable populations [42].

This study has a few limitations. First, daily activities and some symptoms are self-reported and depend on participants’ ability to recall. To minimize this issue, participants were given a computer tablet accompanied by a user-friendly app to report daily symptoms. Since participants were not entirely aware of their air pollution exposures and that this study is prospective in nature, recall bias is not expected to be significant. There may also be concerns that the daily activities during the monitoring period may not reflect the normal daily activities. A cohort study has shown that 71% of participants reported that their activity while wearing the monitor is representative of normal activity [43]. Our study is also limited by the low sample size, especially for the GPS monitoring portion, leading to limited generalizability and greater chance of type II errors. In addition, emerging real-time sensors used in this study do not have the same data quality of the EPA reference monitors although participants mostly spent their time in indoor environments where the sensors perform better than in outdoor environments. We also note that due to the small sample size, we were not able to sufficiently estimate exposure by modelling as a method of assessment, nor perform sensitivity analysis excluding those with compliance below certain thresholds. Lastly, the proportion of waking-time during which personal monitors were worn was at most moderate, which potentially limits the generalizability and validity of personal monitoring. Nevertheless, this is the first study to compare air pollution exposures using different assessment strategies including personal monitoring among women with and without asthma. Our findings add to the limited literature on air pollution exposures during pregnancy and highlight the need for larger studies to understand air pollution exposures and their effects during pregnancy. Given the adverse outcomes of pregnancy previously shown to be associated with ambient air pollution, including pregnancy loss [14], stillbirth [11], preterm birth, and neonatal complications [6,7,10,12,13], improving exposure assessment is a key component to strategies for improving health outcomes for this vulnerable population.

## 5. Conclusions

Pregnant women with and without asthma appeared to be exposed to the same level of air pollution when measured by personal or indoor air monitors. However, exposure profiles differed significantly by assessment method. Local outdoor monitor methods appeared to underestimate PM_2.5_ and ambient temperature, and overestimate O_3_ and NO_2_ exposures relative to personal and indoor monitors. Personal monitors worn throughout the day may best reflect true exposure and are feasible but compliance with continual use was not high, making indoor monitors a potentially more feasible alternative. Furthermore, given that pregnant women who reported certain daily activities (e.g., smoking and using a gas range) had higher air pollution exposures compared to those who did not, it is important to raise awareness on the health effects of air pollution, and strategies that can minimize exposures for pregnant women. These may include using vent hoods during cooking, taking advantage of air purifiers when necessary, and avoiding active and passive smoking. Improving exposure assessment will also allow us to better understand the effects of air pollution during pregnancy. Effective strategies to minimize air pollution exposures among pregnant women are warranted and may facilitate physician involvement in clinical translation to mitigate these potential risks for adverse outcomes among pregnant women with and without asthma.

## Figures and Tables

**Figure 1 ijerph-17-04888-f001:**
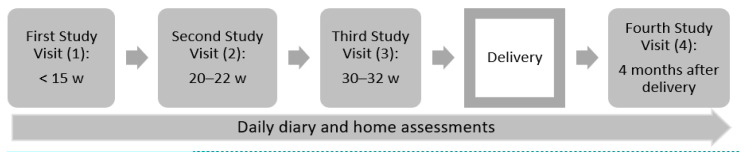
Schematic Timeline of B-WELL-MOM Study Visits.

**Table 1 ijerph-17-04888-t001:** Characteristics of B-WELL-MOM participants who participated in the air-monitoring sub-study (*n* = 40).

Characteristics	*n* (%) or Mean (SD)				
	Overall (*n* = 40)	No Asthma (*n* = 9, 22.5%)	Well Controlled (*n* = 19, 47.5%)	Poorly Controlled (*n* = 12, 30.0%)	*p* ^a^
Study site (n, %)					
Site 1	36 (90.0)	8 (88.9)	18 (94.7)	10 (83.3)	0.67
Site 2	4 (10.0)	1 (11.1)	1 (5.3)	2 (16.7)	
Gestational age at enrollment (week, mean, SD)	24.0 (4.6)	27.1 (5.5)	23.1 (4.5)	23.1 (2.8)	0.06
Maternal age group (year, n, %)					
<23	1 (2.5)	1 (11.1)	0 (0)	0 (0)	0.42
23–29	10 (25.0)	1 (11.1)	5 (26.3)	4 (33.3)	
≥30	29 (72.5)	7 (77.8)	14 (73.7)	8 (66.7)	
Maternal body mass index (kg/m^2^, n, %)					
Normal	8 (20.0)	1 (11.1)	5 (26.3)	2 (16.7)	0.68
Overweight/obese	32 (80.0)	8 (88.9)	14 (73.7)	10 (83.3)	
Monitor-wearing time (%, mean, range)	38.4 (16.5–85.2)	39.2 (16.6–69.2)	40.4 (16.5–77.1)	44.9 (24.3–85.2)	0.73
Waking-hour monitor-wearing time (%, mean, range)	55.7(22.8–100.0)	59.6(22.8–100)	53.9(24–100)	55.5(30.8–100)	0.85
GPS monitoring					
Yes	24 (60.0)	5 (55.6)	13 (68.2)	6 (50.0)	0.64
No	16 (40.0)	4 (44.4)	6 (31.6)	6 (50.0)	
Reported time spent outdoors per day (mins, mean, range)	93.3 (0–480)	94.2(0–360)	98.6(0–480)	83.1(0–300)	0.67
Average home distance from nearest EPA monitor (km, mean, range)					
PM_2.5_	6.6 (0.67–14.0)	6.2(4.1–8.6)	6.8(0.7–14)	6.7(4.2–9.5)	0.97
Temperature	23.5 (4.1–34.7)	18.7(4.1–29.2)	23.1(11.2–29.3)	28.2(21.4–34.7)	0.09
Ozone	8.5 (2.3–13.6)	8.6(5.3–11.8)	8.7(2.3–14.6)	8.2(4.2–11.9)	0.96
NO_2_	6.3(0.75–14.0)	6.6(0.8–13.4)	7.1(1.0–14.0)	4.3(1.8–8.3)	0.38
Average GPS distance from nearest EPA monitor (km, mean, range)					
PM_2.5_	6.9 (0.38–43.8)	6.3(0.4–17.1)	8.1(0.5–43.8)	5.8(0.7–9.5)	<0.01
Temperature	25.6(0.38–63.5)	19.1(0.4–63.5)	24.7(2.1–30.9)	30.8(12.6–42.3)	<0.01
Ozone	8.3 (0.38–25.2)	8.1(0.4–25.2)	9.3(0.8–25.1)	7.2(1.4–13.9)	<0.01
NO_2_	6.3(0.36–58.2)	8.1(0.4–58.2)	7(0.6–55.9)	4.3(0.7–13.9)	<0.01
Exhaled nitric oxide (ppb, mean, range)	17.6 (5–52)	21.5 (5–44)	20 (7–48)	15.1 (5–52)	0.29
Max peak flow morning (L/min, mean, range)	377.3 (130–650)	406.3 (150–650)	418 (260–540)	358 (160–550)	<0.01
Max peak flow afternoon (L/min, mean, range)	340.4 (100–540)	384.8 (140–540)	379.4 (210–500)	323.2 (100–510)	<0.01
Rate of events (per 100 person-day reported, mean, range)					
Wheeze	7.3 (0–100)	0 (0–0)	3.1 (0–33.3)	19.6 (0–100)	<0.01
Cough	27.2 (0–100)	30.6 (0–100)	27.9 (0–100)	23.4 (0–75.0)	0.10
Shortness of breath	31.5 (0–100)	28.7 (0–100)	26.9 (0–100)	41.0 (0–100)	0.20
Chest tightness	7.1 (0–100)	4.0 (0–25.0)	10.8 (0–100)	3.7 (0–25)	0.09
Chest pain	2 (0–25)	5.0 (0–25.0)	1.1 (0–20.0)	1.2 (0–14.3)	0.38
Nausea	17 (0–100)	20.8 (0–75.0)	12.8 (0–80.0)	20.7 (0–100)	0.65
Runny nose	47.5 (0–100)	33.3 (0–100)	56.1 (0–100)	44.5 (0–100)	<0.01
Missed work	3.5 (0–60)	2.8 (0–25.0)	5.3 (0–60.0)	1.2 (0–14.3)	0.45
Wake up at night	14.8 (0–100)	33.6 (0–100)	4.2 (0–80.0)	17.7 (0–100)	0.07
Exercise	21.1 (0–100)	12.9 (0–40.0)	22.5 (0–100)	25.1 (0–100)	0.49
Smoke	2.5 (0–100)	0 (0–0)	0 (0–0)	8.3 (0–100)	0.31
Around a smoker	10 (0–100)	16.1 (0–100)	5.4 (0–66.7.0)	12.8 (0–100)	0.36

GPS, global positioning system; EPA, US Environmental Protection Agency; PM_2.5_, particulate matter <2.5 microns; NO_2_, nitrogen dioxide; ^a^
*p*-values are for differences in characteristics across asthma status and were obtained from exact tests for categorical variables, and mixed linear models for continuous variables to account for within-woman variation.

**Table 2 ijerph-17-04888-t002:** Distribution of air pollution exposure by asthma status.

Pollutant	Assessment Method ^a^	Mean (Min–Max)				
		All	No Asthma	Well Controlled	Poorly Controlled	*p*-Value ^b^
PM_2.5_ (µg/m^3^)	Home + EPA monitor	9.1 (1.7–28.5)	9.7 (3.6–19)	8.5 (1.7–16)	9.2 (3.6–28.5)	0.35
	GPS + EPA monitor	9.3 (1.7–28.5)	10.3 (3.6–28)	8.4 (1.7–14.6)	9.2 (3.6–28.5)	0.11
	Indoor	14.2 (1–132.8)	12.5 (2.3–76.5)	15.8 (1–132.8)	12.8 (2.5–85.8)	0.69
	Personal	26.7 (0.9–665.6)	15.1 (2.1–122.3)	42.2 (0.9–665.6)	12.1 (2.4–44.7)	0.17
	*p*-value exposure ^c^	0.001	0.42	0.01	0.22	
Ozone (ppb)	Home + EPA monitor	28.9 (5.3–68.8)	24.6 (5.3–65)	28.4 (5.4–59.5)	33.9 (6.3–68.8)	0.01
	GPS + EPA monitor	28.9 (5.3–68.8)	24.6 (5.3–65)	28.5 (5.4–59.5)	33.9 (6.3–68.8)	0.01
	Indoor	2.6 (1.1–23.9)	2.2 (1.6–3)	3.2 (1.1–23.9)	2.1 (1.1–3.2)	0.16
	Personal	3.1 (1.5–12.2)	3.5 (1.9–7.6)	2.7 (1.9–5.5)	3.4 (1.5–12.2)	0.13
	*p*-value exposure ^c^	<.0001	<0.0001	<0.0001	<0.0001	
NO_2_ (ppb)	Home + EPA monitor	16.3 (4–35.8)	17.3 (6.9–31)	15.7 (5.9–35.8)	16.5 (4–35)	0.42
	GPS + EPA monitor	16.5 (4–35)	18.3 (7.9–31.6)	15.6 (5.5–34.7)	16.4 (4–35)	0.07
	Indoor	4.6 (0–39)	3.5 (0–15.8)	5.1 (0–27.9)	4.7 (0–39)	0.66
	Personal	5.1 (0–27.4)	4.8 (0–17.4)	5.6 (0–27.4)	4.5 (0–23.1)	0.67
	*p*-value exposure ^c^	<0.0001	<0.0001	<0.0001	<0.001	
Temperature (°C)	Home + EPA monitor	12.6 (−16.3–28.8)	9.8 (−4.4–28.5)	12.1 (−16.3–28.8)	15.9 (−4.8–27)	0.01
	GPS + EPA monitor	12.6 (−16.3–28.8)	9.7 (−4.4–27.8)	12 (−16.3–28.8)	15.9 (−4.8–27)	0.01
	Indoor	24.3 (17.1–52.8)	25 (19.5–31.7)	24.4 (17.1–52.8)	23.7 (18.1–31)	0.38
	Personal	23.6 (13.1–38.7)	24.5 (13.1–38.7)	23.6 (16.8–33.7)	23 (16–30.3)	0.36
	*p*-value exposure ^c^	<0.0001	<0.0001	<0.001	<0.001	

GPS, global positioning system; EPA, US Environmental Protection Agency; PM_2.5_, particulate matter <2.5 microns; NO_2_, nitrogen dioxide; ^a^ The GPS and home method includes 24 participants, the personal monitoring method analysis includes 39 participants, and the indoors method includes 40 participants. ^b^
*p*-values were obtained for comparison across asthma status by mixed models to account for within-person variation. ^c^
*p*-values were obtained for comparison across assessment methods by mixed models to account for within-person variation.

**Table 3 ijerph-17-04888-t003:** Percent difference (and 95% confidence intervals) in air pollutants associated with specific daily activities.

Activities (Yes vs. No)	PM_2.5_ (μg/m^3^)		NO_2_ (ppb)		Ozone (ppb)	
	Personal	Indoors	Personal	Indoors	Personal	Indoors
Near heater	−43.0 (−88.7, 187.6)	-	170.9 (−91.0, 8040.1)	-	20.7 (−33.6, 119.2)	-
Near burning trash	28.8 (−69.0, 435.1)	68.3 (−54.7, 524.8)	28.8 (−91.4, 1831.2)	276.6 (−91.2, 16,070.4)	−8.2 (−45.7, 55.1)	11.7 (−40.5, 109.8)
Used gas appliance	46.4 (−21.3, 172.5)	194.6 (55.0, 459.6) *	33.2 (−53.5, 281.4)	92.4 (−72.9, 1268.3)	−17.1 (−34, 4.0)	−3.1 (−35.5, 45.7)
Used clothes dryer	11.6 (−48.1, 140.0)	126.1 (−20.3, 541.4)	58.2 (−70.6, 752.3)	31.6 (−92.3, 2133.9)	0.7 (−24.5, 34.3)	10.9 (−34.8, 88.5)
Near water heater	−11.3 (−99.4, 12,893.3)	68.4 (−61.5, 636.1)	52.1 (−99.9, 209,422.1)	549.8 (−60.6, 10,626.8)	51.4 (−50.3, 361.0)	40.8 (−25.5, 166.2)
Used printer	4.0 (−46.3, 101.5)	-	−26.9 (−77.1, 133.7)	-	−19.4 (−36.5, 2.3)	-
Smoking	88.2 (−83.5, 2048)	398.7 (78.5, 1293.8) *	66.9 (−95.8, 6499.9)	1008.9 (−67.5, 37,779.1)	47.4 (−5.5, 130.0)	12.3 (−40.2, 110.8)
Near smoker	29.9 (−87.8, 1283.9)	3.1 (−67.7, 229.1)	−78.0 (−99.5, 844.9)	18.5 (−99.3, 20,321.0)	54.1 (−15.4, 180.7)	−0.2 (−33.7, 50.1)
Exercise	−1.6 (−47.1, 83.2)	36.3 (−25.8, 150.3)	−0.6 (−56.2, 125.5)	−19.2 (−78.5, 203.1)	1.8 (−13.8, 20.1)	8.7 (−12.2, 34.6)

PM_2.5_, particulate matter <2.5 microns; NO_2_, nitrogen dioxide; Blank cell indicates that the model did not converge due to small numbers; Linear mixed models adjusted for within-person variation; air pollutants were log transformed; * Indicates statistical significance at alpha < 0.05.

**Table 4 ijerph-17-04888-t004:** Factors associated with compliance (proportion of waking-hour monitor-wearing time).

Characteristics	β (95% CI)	*p* ^a^
Asthma status		
No asthma	reference	
Well controlled	−4.23 (−23.1, 14.64)	0.66
Poorly controlled	−4.68 (−26.15, 16.78)	0.67
Site		
Site 1	reference	
Site 2	11.73 (−8.08, 31.54)	0.25
Maternal age group (years)		
<23	reference	
23–29	7.60 (−7.93, 23.12)	0.34
≥30	7.48 (−1.42, 16.38)	0.10
Maternal body mass index		
Normal	reference	
Overweight/obese	−9.6 (−26.77, 7.57)	0.27
Exhaled nitric oxide (ppb)	0.1 (−0.48, 0.68)	0.74
Max morning peak flow (L/min)	−0.01 (−0.06, 0.04)	0.66
Max afternoon peak flow (L/min)	−0.02 (−0.07, 0.03)	0.45
Symptoms (yes vs. no)		
Wheeze	−5.66 (−14.94, 3.62)	0.23
Cough	0.79 (−13.34, 14.93)	0.91
Shortness of breath	−2.99 (−14.16, 8.18)	0.60
Chest tightness	9.88 (−10.92, 30.67)	0.35
Chest pain	9.29 (−6.76, 25.34)	0.26
Nausea	13.27 (−2, 28.54)	0.09
Runny nose	2.06 (−9.57, 13.7)	0.73
Missed work	19.81 (0.91, 38.72)	0.04
Wake up at night	3.17 (−15.87, 22.2)	0.74
Exercise	−10.64 (−18.66, −2.61)	0.01
Smoke	−22.83 (−29.94, −15.71)	<0.001
Around a smoker	−16.24 (−26.43, −6.04)	<0.001
Used prescription drug	−13.39 (−26.42, −0.37)	0.04
Used over the counter drug	−10.37 (−26.39, 5.66)	0.20
Near heater	42.05 (33.8, 50.31)	<0.001
Near burning trash	−7.33 (−22.81, 8.16)	0.35
Used gas appliance	−8.74 (−22.99, 5.5)	0.23
Used clothes dryer	−6.29 (−20.58, 8)	0.39
Near water heater	15.41 (−6.13, 36.95)	0.16
Used printer	−1.11 (−15.94, 13.72)	0.88
Times spent outdoors (minutes)	0.03 (−0.04, 0.1)	0.38

^a^ Linear mixed models adjusted for within-person variation.

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
