# Peer review of "Air Pollution Exposure Monitoring among Pregnant Women with and without Asthma"

_ijerph, 2020, doi:10.3390/ijerph17134888_

Round 1

Reviewer 1 Report

This manuscript shares the findings from an exposure study that aimed to assess personal exposures to air pollution using four different methods.  Although personal monitoring of air pollution is not novel, this study's finding are interesting because of the subjects. 

The authors have provided no details about the monitor calibration or quality control methods used, nor did they provide any explanation as to how the personal exposure estimates were made. It is very unclear as to how any of the measurements were taken as well as the accuracy and precision of the measurements. Did they include any control, or duplicates? Did the use the mass filter option of the MicroPEMS? They also provided no details as to how they handled the 55% compliance of the RTI MicroPEMS. It is almost impossible to review an exposure study manuscript without these basic details. 

Author Response

We appreciate the reviewer’s careful assessment and thoughtful comments on our manuscript titled “Air Pollution Exposure Monitoring Among Pregnant Women with and Without Asthma”. We have carefully addressed these comments in the revised manuscript, and in the one-by-one responses below. 

REVIEWER 1

Comment 1. This manuscript shares the findings from an exposure study that aimed to assess personal exposures to air pollution using four different methods.  Although personal monitoring of air pollution is not novel, this study's finding are interesting because of the subjects.

The authors have provided no details about the monitor calibration or quality control methods used, nor did they provide any explanation as to how the personal exposure estimates were made. It is very unclear as to how any of the measurements were taken as well as the accuracy and precision of the measurements. Did they include any control, or duplicates? Did they use the mass filter option of the MicroPEMS? They also provided no details as to how they handled the 55% compliance of the RTI MicroPEMS. It is almost impossible to review an exposure study manuscript without these basic details.

Authors’ response:  Thank you for this helpful comment. The sensor devices were calibrated and validated before deployment, and PM2.5 concentrations were post-corrected using the MicroPEM mass filter data. In the revised manuscript, we added a section describing the sensor calibration process prior to deployment, other quality control measures, and how personal estimates were measured (Methods, Air Pollution Assessment). We also added a reference with more details about the calibration process for the MicroPEM.

Prior to each MicroPEM deployment, each individual MicroPEM’s flow rate and baseline response were calibrated.23 The real-time PM2.5 concentration data were post-corrected using the corresponding gravimetrically determined concentration measured by the filter samples (gold-standard measurement). Real-time sensor operating parameters such as inlet pressure drop, orifice pressure drop, and flow rate were examined during the validity check. The MicroPEM units were set to run for at least 5 days continuously using 30-second on/off cycling at the flow rate of 0.5 L/min. All Cairclips were examined for precision through a collocation study before the air monitoring study began. For Ogawa badges, shipping and lot blank samples were collected throughout the study to correct for potential contamination and background levels, respectively. (lines 132-141)

Real-time sensors (MicroPEMs, Cairclips) were operated continuously throughout the monitoring period, and each of two Ogawa badges was deployed to collect separate personal and indoor samples. (lines 144-146)

We did not adjust for the 56% monitoring compliance for a few reasons. First, one of the main goals of the study was to investigate the feasibility of personal monitoring in pregnancy by assessing compliance and factors associated with compliance. Second, we also estimated exposures using three other commonly used methods. Third, a prior study showed that compliance does not appear to misrepresent exposures measured longitudinally when one can reach a threshold of about 40% [28]. Lastly, due to the small sample size, we felt representing the data available was preferable to imputing 46% of the monitoring times. The small sample size also challenges the interpretation of excluding participants below certain compliance thresholds, but a larger study in the future would allow such sensitivity analysis. We discuss non-compliance as a potential limitation in our Discussion (lines 353-354):

We also note that due to the small sample size, we were not able to sufficiently estimate exposure by modelling as a method of assessment, nor perform sensitivity analysis excluding those with compliance below certain thresholds. Lastly, the proportion of waking time monitors were worn was at most moderate, which potentially limits the generalizability and validity of personal monitoring.  

Reviewer 2 Report

An interesting research, hard to be conducted.

Plenty of experimental data, which can be of interest for other researchers.

Valuable conclusion for the effect of the assessment method on the exposure profiles.

I miss more practically oriented suggestions and conclusions for both research audience and society.

Author Response

We appreciate the reviewers’ careful assessment and thoughtful comments on our manuscript titled “Air Pollution Exposure Monitoring Among Pregnant Women with and Without Asthma”. We have carefully addressed these comments in the revised manuscript, and in the one-by-one responses below. 

REVIEWER 2

Comment 1: An interesting research, hard to be conducted. Plenty of experimental data, which can be of interest for other researchers. Valuable conclusion for the effect of the assessment method on the exposure profiles. I miss more practically oriented suggestions and conclusions for both research audience and society.

Authors’ response: Thank you for the encouraging and constructive comments. As the reviewer noted, our findings have important implications for both the research community and the general public. From the research perspective, our findings provided useful information on a) how common air pollution assessment methods differ, and b) monitor wearing compliance in personal monitoring during pregnancy. These findings can inform future studies regarding how to approach exposure assessment for air pollution during pregnancy. From a more practical perspective, our findings provide empirical evidence in terms of which daily activities may expose pregnant women with and without asthma to higher pollution.

We described these implications throughout the Discussion section. From the research perspective, we offered a discussion regarding which assessment method may be more feasible for future studies (lines 297-303):

Our data also show that exposures varied significantly by assessment methods. More specifically, local outdoor monitor-based (e.g., EPA monitor) methods appeared to underestimate PM2.5 and ambient temperature, and overestimate O3 and NO2 exposures relative to personal and indoor monitors. These differences have important implications for future studies as they support the notion that local air monitors lack the ability to capture small spatial variations in different microenvironments in exposure.

We also mentioned (Line 306-308):

Our data also show that the GPS-based and home-based methods yielded similar estimates with high correlation, suggesting that accounting for outdoor space-time activities do not significantly influence exposure estimates 35.

Finally, we recommended that (lines 315-318):

Given women spend most of their times indoors 38, a trend that is also reflected in our data, indoor monitoring may offer a reasonable alternative to alleviate participant burden, especially considering our results which suggest that indoor estimates are very similar to measurements from personal monitors.

In the revised manuscript, we added additional practical/societal recommendations that can be derived from our findings (Discussion, lines 325-331).

...some common daily activities, including use of gas appliances, are associated with higher exposure to pollutants. As such, it is important to raise awareness regarding the potentially harmful effects of these pollutants and encourage strategies for minimizing exposure such as the use of vent hoods, air purifiers, and the avoidance of active/passive smoking. Meanwhile, there is a need for more comprehensive and larger studies to investigate personal exposure data together with daily diaries to identify activities that cause high exposure conditions in order to prepare effective and reasonable intervention efforts.

In the conclusion, we also added (lines 372-376):

Furthermore, given pregnant women who reported certain daily activities (e.g., smoking, using gas range) had higher air pollution exposures compared to those who did not, it is important to raise awareness on the health effects of air pollution, and strategies that can minimize exposures for pregnant women. These may include using vent hoods during cooking, taking advantage of air purifiers when necessary, and avoiding active and passive smoking.

Reviewer 3 Report

This is a well-written paper. However, I wished that there was an emphasis on the side effects of exposures of pregnant women to air pollution. The study should not simply be testing the exposures of pregnant women to air pollution, rather, the implication for their health and that of their children. Perhaps, there will be a follow-up study to cater to that.

In your conclusion, you made mention of exposure to air pollution being high during pregnancy depending on activities. I do not think that should be. Although, the pregnancy period is a unique period for women, I do not think that being pregnant necessarily increases exposure to air pollution. However, whatever the case may be, I can see that so much efforts were put into the work, it was thorough in terms of data collection, analysis and writing. However, if possible, I wish to read more about how this translates to decision making with regards to air pollution exposure during pregnancy.

Author Response

We appreciate the reviewers’ careful assessment and thoughtful comments on our manuscript titled “Air Pollution Exposure Monitoring Among Pregnant Women with and Without Asthma”. We have carefully addressed these comments in the revised manuscript, and in the one-by-one responses below. 

REVIEWER 3

Comment 1. This is a well-written paper. However, I wished that there was an emphasis on the side effects of exposures of pregnant women to air pollution. The study should not simply be testing the exposures of pregnant women to air pollution, rather, the implication for their health and that of their children. Perhaps, there will be a follow-up study to cater to that.

Authors’ response:  We agree with the reviewer regarding the importance of assessing the health effects of air pollution in pregnant women. Such questions will be explored in future analyses, but a detailed assessment of health effects is beyond the scope of this study. Our objectives were to investigate air pollution exposures among women with and without asthma, compare assessment methods, and investigate compliance to personal monitoring. The study objectives are specified both the Introduction and Discussion sections.

We added more context on the adverse outcomes of pregnancy associated with air pollution exposure among asthmatic women to the introduction (lines 55-59)

Ubiquitous environmental exposures, such as air pollution, increase the risk of asthma exacerbation in the general population through oxidative stress and inflammatory responses 4,5. However, the effects of common air pollutants on asthma among pregnant women have not been well-investigated. Asthma has been associated with higher risk for preeclampsia6 and preterm birth7 as well as neonatal respiratory complications associated with air pollutants but not all adverse outcomes of pregnancy vary by asthma status.8,9 Regardless of asthma status, air pollution exposure appears to increase risk for adverse pregnancy outcomes including pregnancy loss, stillbirth, preterm delivery and infant growth restriction.10-14

In the Discussion, we stressed the importance of future studies evaluating health effects of air pollution exposure for pregnant women (lines 358-363): 

Our findings add to the limited literature on air pollution exposures during pregnancy and highlight the need for larger studies to understand air pollution exposures and their effects during pregnancy. Given the adverse outcomes of pregnancy previously shown to associate with ambient air pollution, including pregnancy loss,14 stillbirth,11 preterm birth, and neonatal complications, 6,7,10,12,13 improving exposure assessment is a key component to strategies for improving health outcomes for this vulnerable population.

In the conclusion, we added (lines 376-380)

Improving exposure assessment will also allow us to better understand the effects of air pollution during pregnancy. Effective strategies to minimize air pollution exposures among pregnant women are warranted and may facilitate physician involvement in clinical translation to mitigate these potential risks for adverse outcomes among pregnant women with and without asthma.

Comment 2. In your conclusion, you made mention of exposure to air pollution being high during pregnancy depending on activities. I do not think that should be. Although, the pregnancy period is a unique period for women, I do not think that being pregnant necessarily increases exposure to air pollution. However, whatever the case may be, I can see that so much efforts were put into the work, it was thorough in terms of data collection, analysis and writing. However, if possible, I wish to read more about how this translates to decision making with regards to air pollution exposure during pregnancy.

Authors’ response: Thank you for the opportunity to clarify. We compared air pollution exposures during pregnancy between women who reported certain activities compared to those who did not. The findings showed that women with certain activities such as smoking and using gas appliance had higher exposures. This is consistent with prior knowledge about the type of pollutants emitted by these activities. In the conclusion, we revised the sentence in question to better reflect that the comparison is not made between pregnant and non-pregnant women, and provided more specific recommendations as suggested (Lines 372-380).

Furthermore, given pregnant women who reported certain daily activities (e.g., smoking, using gas range) had higher air pollution exposures compared to those who did not, it is important to raise awareness on the health effects of air pollution, and strategies that can minimize exposures for pregnant women. These may include using vent hoods during cooking, taking advantage of air purifiers when necessary, and avoiding active and passive smoking. Improving exposure assessment will also allow us to better understand the effects of air pollution during pregnancy. Effective strategies to minimize air pollution exposures among pregnant women are warranted and may facilitate physician involvement in clinical translation to mitigate these potential risks for adverse outcomes among pregnant women with and without asthma.

A similar note was also added to the Discussion (lines 326-329):

As such, it is important to raise awareness regarding the potentially harmful effects of these pollutants and encourage strategies for minimizing exposure such as the use of vent hoods, air purifiers, and the avoidance of active/passive smoking.